



# Implementation of an Ensemble Kalman Filter in the Community Multiscale Air Quality Model (CMAQ Model v5.1) for Data Assimilation of Ground-level PM$_{2.5}$

Soon-Young Park[1,2], Uzzal Kumar Dash[1], Jinhyeok Yu[1], Keiya Yumimoto[3], Itsushi Uno[3], and Chul Han Song[1]

[1]School of Earth Sciences and Environmental Engineering, Gwangju Institute of Science and Technology (GIST), Gwangju, 61005, Republic of Korea
[2]Institute of Environmental Studies, Pusan National University, Busan, 46241, Republic of Korea
[3]Research Institute for Applied Mechanics, Kyushu University, Fukuoka, 816-8580, Japan

*Correspondence to*: Chul Han Song (chsong@gist.ac.kr)

**Abstract.** In this study, we developed a data assimilation (DA) system for chemical transport model (CTM) simulations using an ensemble Kalman filter (EnKF) technique. This DA technique is easy to implement to an existing system without seriously modifying the original CTM, and can provide flow-dependent corrections based on error covariance by short-term ensemble propagations. First, the PM$_{2.5}$ observations at ground stations were assimilated in this DA system every 6 hours over South Korea for the period of the KORUS–AQ campaign, from 1 May to 12 June, 2016. The DA performances with the EnKF were then compared to a control run (CTR) without DA, as well as a run with three-dimensional variational (3DVAR) DA. Consistent improvements due to the ICs assimilated with the EnKF were found in the DA experiments with 6 h interval, compared to the CTR run, and to the run with 3DVAR. In addition, we attempted to assimilate the ground observations from China to examine the impacts of improved boundary concentrations (BCs) on the PM$_{2.5}$ predictability over South Korea. The contributions of the ICs and BCs to improvements in the PM$_{2.5}$ predictability were also quantified. For example, the relative reductions in terms of the normalized mean bias (NMB) were found to be about 27.2 % for the 6 h reanalysis run. A series of 24 hour PM$_{2.5}$ predictions were additionally conducted each day at 00 UTC with the optimized initial concentrations (ICs). The relative reduction of the NMB was 17.3 % for the 24 h prediction run, when the updated ICs were applied at 00 UTC. This means that after the application of the updated BCs, an additional 9.0 % reduction in the NMB was achieved for 24 h PM$_{2.5}$ predictions in South Korea.

## 1 Introduction

Among many air pollutants, particular attention has been paid to the issue of atmospheric aerosols in East Asia and South Korea, where large anthropogenic emissions from growing economic activities cause frequent high episodes of air pollution. Several environmental and epidemiological studies have suggested that continual exposure of particulate matter with aerodynamic diameter smaller than 2.5 μm (PM$_{2.5}$) has critical effects on human mortality and morbidity (Pope and Dockery,



2006; Cohen et al., 2017; Dehghani et al., 2017). Because of the severity of the influences of PM$_{2.5}$ on human health, the accuracy of PM$_{2.5}$ forecasts has become a central issue in South Korea. To achieve the goal of improving PM$_{2.5}$ predictability, the National Institute of Environmental Research (NIER) of South Korea has implemented daily operational air quality forecast since 2014, using the 3-D Chemical Transport Model (CTM) (Chang et al., 2016), while the Korean Ministry of the

Environment (KMoE) provides real-time observations of PM$_{2.5}$, together with the concentrations of five other criteria air pollutants (PM$_{10}$, O$_3$, CO, SO$_2$, and NO$_2$) in a website named "Air Korea" (https://www.airkorea.or.kr). Although in general, the CTM simulation can overcome the spatial and temporal limitations of ground observations, it has large uncertainties that are due to imperfect emissions, initial conditions (ICs), boundary conditions (BCs), meteorological fields, and physical and photo-chemical mechanisms (Carmichael et al., 2008; Solazzo et al., 2012).

To improve the accuracy of the short-term predictions via the CTM simulations, chemical data assimilation (DA) has been proposed as an effective method to reduce the uncertainties in the CTM parameters (e.g., Sandu and Chai, 2011; Zhang et al., 2012b, a; Bocquet et al., 2015; Menut and Bessagnet, 2019). The chemical DA is a technique for integrating information provided by noisy observations and imperfect background estimations from CTM simulations. This integration of the two groups of information can theoretically better represent the true state of the chemical atmosphere. The DA techniques have

been predominantly applied in the Numerical Weather Prediction (NWP) (Kalnay, 2002), such as Optimal Interpolation (OI: Lorenc, 1981); three-dimensional variational method (3DVAR: Lorenc, 1986; Parrish and Derber, 1992; Rabier et al., 1998); four-dimensional variational method (4DVAR: Talagrand and Courtier, 1987; Courtier et al., 1994; Rabier et al., 2000); and Ensemble Kalman Filter (EnKF: Evensen, 2003). While the utilization of DA techniques in air quality predictions has been limited, these techniques have more recently started to be used for air quality prediction, as well. To date, several DA

methods have been applied to optimize the uncertainties in model input parameters, including ICs (e.g., Elbern and Schmidt, 2001; Park et al., 2016), BCs (e.g., Roustan and Bocquet, 2006), and emissions fluxes (e.g., Elbern et al., 2007).

For the past two decades, various DA algorithms have been applied, especially to aerosol prediction studies. Several studies have focused on assimilating aerosol observations via OI (Lee et al., 2013; Park et al., 2011; Park et al., 2014; Tang et al., 2015; Tang et al., 2017; Chai et al., 2017; Lee et al., 2020a); 3DVAR (Pagowski et al., 2010; Liu et al., 2011; Schwartz et al.,

2012; Saide et al., 2013; Jiang et al., 2013; Li et al., 2013; Pang et al., 2018; Ha et al., 2020); and 4DVAR (Benedetti et al., 2019; Morcrette et al., 2009). All the previous studies mentioned above have reported that the OI, 3DVAR, and 4DVAR assimilations using satellite-retrieved or ground-based observations led to improved aerosol predictability.

Even so, each of these DA methods has its own limitations. The OI and 3DVAR usually employ isotropic corrections due to a static (i.e., time-invariant) background error covariance (BEC), based on model climatological profiles. Although the

4DVAR has been reported to show better performance than the OI and 3DVAR, it requires constant development and maintenance of a tangent linear and adjoint model, which may be a time-consuming and labor-intensive task (Skachko et al., 2014). On the other hand, the EnKF is relatively easy to implement without requiring a tangent linear or adjoint model, and can easily compute flow-dependent BEC from short-term ensemble predictions. This flow dependence of the BEC is one of the main reasons behind the possible success of the EnKF method, compared to other DA methods. Several studies (Tang et





al., 2011; Pagowski and Grell, 2012; Yumimoto and Takemura, 2015; Rubin et al., 2016; Yumimoto et al., 2016; Peng et al., 2017; Peng et al., 2018; Lopez-Restrepo et al., 2020) applied the EnKF DA approach to improve the accuracy of air quality prediction via assimilating surface and/or satellite observations. For example, Yumimoto et al. (2016) conducted the application of the EnKF method with satellite-retrieved aerosol observations to evaluate the effectiveness of the DA on dust forecast, and found improved agreement between the predictions and observations. More recently, Peng et al. (2017) reported significant improvements in $PM_{2.5}$ prediction via the joint optimization of ICs and emissions using an EnKF method, assimilating ground-based $PM_{2.5}$.

To optimize the ICs, two studies (Lin et al., 2008; Candiani et al., 2013) carried out assimilation using ground-based aerosol observations with different variants of EnKF DA algorithms. However, few studies have applied the EnKF method, examining the importance of the BCs. When long-range transport is an important issue, the BCs can be an important information. For example, Constantinescu et al. (2007b) and Constantinescu et al. (2007a) extended the EnKF method in a direction to consider the lateral boundary conditions, and to correct emission flux factors in the assimilation process by solving the state parameter estimation problem. Other than this study, no prior study has applied the EnKF method to this type of research, particularly with the Community Multiscale Air Quality (CMAQ) model.

This work is a new endeavor to develop an EnKF DA system for the CMAQ model. The period of the KORUS–AQ campaign 2016 (1 May to 11 June, 2016) was chosen to be the target period to test the developed EnKF DA system, since this period includes well-defined and various types of air pollution episodes, e.g., Yellow dust event, stagnant high PM episode, long-range transport events, and rainy days (Peterson et al., 2019; Jordan et al., 2020). To improve the predictability of $PM_{2.5}$ in South Korea for this period, ground-based $PM_{2.5}$ data were assimilated to update the IC and BC fields. Since this is our first attempt to develop an EnKF DA system, we also compared the performances of the EnKF DA system with the existing 3DVAR DA algorithm (Lee et al., 2020, in preparation).

We believe that this study can be distinguishable from other EnKF studies in three aspects: (i) The EnKF chemical DA system was first developed to assimilate $PM_{2.5}$ for/with the CMAQ model. In particular, this study intended to enhance the accuracy of the $PM_{2.5}$ prediction via assimilating the ground-observed $PM_{2.5}$ in South Korea (nearly 150 stations) and China (nearly 850 stations). The advantages of the assimilation of the ground-observed $PM_{2.5}$ are also discussed in the text. (ii) The first developed EnKF DA system was applied to the $PM_{2.5}$ predictions in South Korea, where air quality is frequently influenced by long-range transport from the Eastern, Northern, and Northeastern parts of China (EC, NC, and NEC in Fig. 1). (iii) To evaluate the influences of inflow from China on air quality in South Korea more quantitatively, this study assimilated the ground observations from China and South Korea separately.

The manuscript is organized as follows. Sections 2 describes the methodology of this study, including the DA algorithm, CTM, observations, and experimental settings. Section 3.1 discusses the effects of assimilation of ground-based observations, and then compares the results with those from 3DVAR, based on the reanalysis results. Section 3.2 provides the results of improved boundary conditions in one-day prediction simulations. Then, Section 3.3 quantifies the contributions of updating ICs and BCs with statistical analysis. Finally, Section 4 concludes the paper.





## 2 Methods

**2.1 Ensemble Kalman Filter (EnKF)**

The EnKF is a DA technique, first introduced by (Evensen, 1994), which was an approximate version of the Kalman filter (KF) (Kalman, 1960). The basic principle of the KF is to estimate a true state, while minimizing the variances of the state with a linear combination of the best estimates of the model and the observations. The optimal state estimated from the KF shows less uncertainty than the model predictions and observations. This optimal state is called the 'analysis'. To apply the

KF to a non-linear model, a tangent linear model needs to be constructed, as well as its adjoint. However, the EnKF requires neither a tangent linear model nor its adjoint, since it employs Monte Carlo approximation that can estimate the model error covariances using finite ensemble simulations (Evensen, 1994). In particular, the model error covariances used in the EnKF technique are flow-dependent, which is the one of the major differences from other DA methods.

The theoretical foundation of the EnKF method proposed by (Evensen, 2003) is briefly presented below:

$$\mathbf{x}_{i,k}^f = \mathcal{M}\mathbf{x}_{i,k-1}^a + \mathbf{q}_{i,k}, \qquad i = 1,2,\dots,N \tag{1}$$

$$\mathbf{y}_{i,k}^o = \mathbf{y}_k^o + \boldsymbol{\epsilon}_{i,k}^o \tag{2}$$

$$\mathbf{x}_{i,k}^a = \mathbf{x}_{i,k}^f + \mathbf{K}_k\big(\mathbf{y}_{i,k}^o - \mathbf{H}\mathbf{x}_{i,k}^f\big) \tag{3}$$

$$\mathbf{K}_k = \mathbf{P}_k^f \mathbf{H}^T \big(\mathbf{H}\mathbf{P}_k^f \mathbf{H}^T + \mathbf{R}_k\big)^{-1} \tag{4}$$

where, the subscripts $i$ and $k$ represent the $i$-th ensemble member and the time sequence, respectively. In this set of equations, the first information to estimate the true state is the forecast state, $\mathbf{x}_{i,k}^f$ in Eq. (1). This is the predicted state estimated from the model simulation ($\mathcal{M}$) using the updated initial state, $\mathbf{x}^a$, of the previous time step ($k-1$). Here, $\mathbf{x}_{i,k-1}^a$ is obtained via DA. The model predictions also include pseudo-random model error, $\mathbf{q}$, drawn from Gaussian probability distribution function (PDF) with zero mean value and covariance, $\mathbf{P}^f$, [$\mathbf{q} \sim N(0, \mathbf{P}^f)$]. The second item of information is the

observations, $\mathbf{y}_i^o$ at time $k$, which are randomly sampled from the PDF of the observations. The PDF of the observations can be generated based on error information of the observed values. Each ensemble member is generated with the assimilation of perturbed observations ($\mathbf{y}_i^o$). The new analyses are then conducted, following Eq. (3). These analyses are used for the next ensemble predictions (we term them 'propagations'). $\mathbf{H}$ is a linear operator that transforms the model space into the observation space. $\mathbf{K}$ is the Kalman gain matrix at a specific time that includes both model and observation errors shown in

Eq. (4). The observation error covariance matrix, $\mathbf{R}$, contains measurement and representation errors, and can be calculated from the defined observation error, $\boldsymbol{\epsilon}^o$, [$\mathbf{R} = \overline{\boldsymbol{\epsilon}^o(\boldsymbol{\epsilon}^o)^T}$]. $\mathbf{P}^f$ is the model error covariance matrix that explains the spatial error correlations and error correlations among the model variables. This can be estimated via the ensemble approach formulated in Eqs. (5) and (6) shown below:

$$\mathbf{P}_k^f \mathbf{H}^T \equiv \frac{1}{N-1} \sum_{i=1}^N \left(\mathbf{x}_{i,k}^f - \overline{\mathbf{x}_k^f}\right)\left(\mathbf{H}\mathbf{x}_{i,k}^f - \overline{\mathbf{H}\mathbf{x}_k^f}\right)^T \tag{5}$$

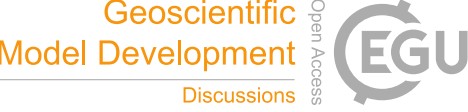

$$\mathbf{HP}_k^f \mathbf{H}^T \equiv \frac{1}{N-1} \sum_{i=1}^{N} \left( \mathbf{Hx}_{i,k}^f - \overline{\mathbf{Hx}_k^f} \right) \left( \mathbf{Hx}_{i,k}^f - \overline{\mathbf{Hx}_k^f} \right)^T \tag{6}$$

where, the overbar represents the ensemble mean. One of the advantages of the EnKF method is that instead of storing a full covariance matrix ($\mathbf{P}^f$), the error statistics can be computed by Eqs. (5) and (6) using ensembles of model states with the assumption that the ensemble mean can be the best estimate of the true state.

The practical approaches to implement Eqs. (1)–(6) are described as follows. First, through multiple pre-sensitivity tests with the considerations of both model performances and computational costs, the total number of the ensembles ($N$) was determined to be 40. Although the results of these sensitivity tests are not presented in this manuscript, this number of the ensemble members (N=40) has generally been used in many other EnKF applications (e.g., Schutgens et al., 2010; Coman et al., 2012; Dai et al., 2014).

Second, the diagonal components in the observation error covariance matrix, $\mathbf{R}$, were calculated based on the assumption that no errors are correlated among observation locations. The components of $\mathbf{R}$ matrix have been estimated, while considering the contributions from measurement and representation errors in several previous studies (e.g., Schwartz et al., 2014; Peng et al., 2017; Chen et al., 2019). The application of this method to the observation data has resulted in average observation errors of around 5 % of the observed values. Therefore for simplicity, in this study the observation errors were set to be 5 % of the observations. To generate perturbed observations ($\mathbf{y}_{i,k}^o$) at specific time in Eq. (2), 40 random samples ($\boldsymbol{\epsilon}_i^o$) were drawn from the Gaussian distribution having 0 mean value and standard deviations of 5 % of the observation values.

Since almost no observation locations exactly match the uniform model grid points, an observation operator, $\mathbf{H}$, is required to interpolate the model grid-point concentrations to the observation locations. Thus, $\mathbf{H}$ was constructed as a form of the matrix having weighting factors proportional to the inverse distances from the four edge-points of the model grid that surrounds the observation location.

Third, the method to generate ensemble spread for the model ($\mathbf{x}_i^f$) is as follows. In the CTM runs, the state vector, $\mathbf{x}$, is propagated from time, $k-1$, to time, $k$. This can be expressed in the following discrete form:

$$\mathbf{x}_i^f(k) = \boldsymbol{\mathcal{M}}\big(\mathbf{x}_i^b(k-1), \boldsymbol{\eta}_i^b(k-1)\big) \tag{7}$$

Here, the superscripts $f$ and $b$ denote forecast and background, respectively; while $\boldsymbol{\mathcal{M}}$ denotes the model dynamic operator. The subscript $k$ representing time in the previous section is replaced by $(k)$ here. The state vector $\mathbf{x}$ defined in our study represents the PM$_{2.5}$ IC to be updated, and $\boldsymbol{\eta}$ represents the model parameters that are perturbed, but not updated through EnKF. This indicates that the multivariate covariances among the aerosol species are not considered. In this study, emissions and BCs were considered as $\boldsymbol{\eta}$. The approaches to generate initial ensembles, emissions, and BCs via random perturbation are described below.

The initial ensembles were created by perturbing the background values of state vector, $\mathbf{x}^b$, at time, t = 0, following the equation below:





$$\mathbf{x}_i^b(0) = \mathbf{x}^b(0) + \delta\mathbf{x}_i(0), \qquad i = 1, 2, \dots, N \tag{8}$$

where, $\delta\mathbf{x}_i$ represents the $N$ number of random samples selected from the Gaussian distribution having zero mean and standard deviations of 50 % of the initial concentration at each corresponding model grid. Following this process, we prepared 40 ensemble members ($N = 40$) for the initial ensemble. These 40 initial conditions propagated with time through CTM ($\mathcal{M}$) with another perturbed parameter ($\boldsymbol{\eta}$).

For perturbing BCs and emission rates, we took time-correlated noise into account to maintain the temporal evolution of those parameters. In addition, avoiding the rapid fluctuations of perturbations is another reason behind the use of time-correlated noise (colored noise). The method of adding colored noise is the same as that described in (Tang et al., 2011):

$$\boldsymbol{\eta}_i^b(k) = \boldsymbol{\eta}^b(k) + \delta\boldsymbol{\eta}_i(k) \tag{9}$$

$$\delta\boldsymbol{\eta}_i(k) = \alpha\delta\boldsymbol{\eta}_i(k-1) + \sqrt{1-\alpha^2}\sigma\omega_i(k-1), \qquad i = 1, 2, \dots, N \tag{10}$$

$$\alpha = \exp(-1/\tau) \tag{11}$$

where, $\boldsymbol{\eta}^b$ is the background emission fields or BCs, $\delta\boldsymbol{\eta}_i$ denotes the random perturbation samples obtained from the previous time step, and $\alpha$ represents the smoothing coefficient that is a function of time decorrelation scale ($\tau$), for which we used 24 h. $\omega_i(k-1)$ is the random sample drawn in the previous time step from the Gaussian distribution having zero mean and standard deviation of one. For the standard deviations ($\sigma$), we used 30 % of boundary inflow concentrations for PM$_{2.5}$, and 50 % of background emission rates.

In theory, an ensemble of infinite model states can provide the most realistic estimations of model error. However, because of the limitations of the computational cost, the ensembles with finite size are used to provide an approximation to the error covariance matrix. The limited ensemble size causes a sampling error. Small ensemble size may lead to underestimation of the prediction error covariances, called 'filter divergence' (Houtekamer and Mitchell, 1998), and makes spurious corrections at regions remote from the observation locations, called 'spurious correlation' (Constantinescu et al., 2007b). To avoid such filter divergence and spurious correlation, we applied covariance inflation and localization, respectively. The Gaspari–Cohn piecewise polynomial (Gaspari and Cohn, 1999) with a horizontal width of 100 km and a vertical width of 2 km was used to prevent the spurious correlation by localizing the model error covariances. In addition, the Relaxation-to-Prior-Spread (RTPS) inflation (Whitaker and Hamill, 2012) method was applied against the filter divergence, by inflating the ensemble spread before and after the DA.

## 2.2 Three-Dimensional Variational Data Assimilation (3DVAR)

An analysis state generated by 3DVAR is obtained by minimization of the cost function shown below:

$$J \equiv \frac{1}{2}\left(\mathbf{x}_k - \mathbf{x}_k^f\right)^T \mathbf{B}^{-1}\left(\mathbf{x}_k - \mathbf{x}_k^f\right) + \frac{1}{2}(\mathbf{H}\mathbf{x}_k - \mathbf{y}_k^o)^T \mathbf{R}_k^{-1}(\mathbf{H}\mathbf{x}_k - \mathbf{y}_k^o). \tag{12}$$

Most of the notations in Eq. (12) are the same as those in Eq. (3), except for the time invariant (static) BEC matrix, **B**. The National Centers for Environmental Prediction (NCEP) Grid-point Statistical Interpolation (GSI) provides the 3DVAR DA



algorithm. Based on the GSI version 3.6 (Shao et al., 2016), Lee et al. (2021, in preparation) modified it, making an interface with the CMAQ model. The National Meteorological Centre (NMC) method (Parrish and Derber, 1992) was used to provide the **B** matrix that contains the standard deviations, as well as the vertical and horizontal length scales of the model errors. In the NMC method, the model errors are approximated from a set of differences between the model predictions with different lengths of time window. We used a total of 42 pairs of 12 and 24 h model predictions for the BEC calculations, following the

method of Schwartz et al. (2012). Lee et al. (2021, in preparation) describes the details of the 3DVAR method, including the minimization algorithm, observation operator, and observation error covariances.

### 2.3 Numerical Models and Input Data

In this study, the EnKF DA algorithm was developed for the Weather Research and Forecasting (WRF)-CMAQ modeling system. The WRF–CMAQ system was run in off-line mode, which means that the CMAQ model runs were made

sequentially after the meteorological fields were generated by the WRF model. This section briefly describes the two numerical models, input fields (e.g., emission and meteorology), simulation domains, and observation data used for the DA.

The WRF version 3.8.1 (Skamarock, 2008) with the Advanced Research WRF (ARW) dynamical core was used to produce meteorological fields for the CMAQ model simulations. The ARW dynamical core employs fully compressible and non-hydrostatic Euler equations, together with Arakawa-C grid staggering. In the WRF simulations, the Final (FNL) operational

global analyses data produced by the NCEP (Saha et al., 2010) were used for the ICs and BCs. Temporal and spatial resolutions of the FNL data are 6 hours and 0.25 degree, respectively. To minimize the uncertainty in the meteorological fields, the ground measurements and vertical radiosonde data were also assimilated with 3 and 6 h intervals, respectively, with the Newtonian relaxation (or nudging) method (Stauffer and Seaman, 1990). The hourly meteorological fields were provided by the WRF model simulations, and they were then converted into CMAQ-ready format via the Meteorology–

Chemistry Interface Processor (MCIP v4.3; Otte and Pleim, 2010). Table 1 summarizes the detailed model configurations of the WRF model simulations.

The CMAQ model v5.1 (Byun and Ching, 1999; Byun and Schere, 2006) was used in this study to simulate the atmospheric photo-chemistries, aerosol dynamics and thermodynamics, and transport of atmospheric species. The CMAQ runs have two domains in accordance with our experimental purposes. The horizontal resolutions of the mother domain (D1) and daughter

domain (D2) are 27 and 9 km, respectively, with 15 vertical layers, while the model top being at 20 km. Tables 2 and 3 list the CMAQ model configurations and the domain specifications used in this study, respectively.

The mother domain (D1) for the CMAQ model simulations covers Northeast Asia including China, the Korean Peninsula, and Japan, and the daughter domain (D2) nested in the D1 targets South Korea (refer Fig. 1). With this nesting configuration, we intended to examine how the BCs provided by the D1 affect the $PM_{2.5}$ predictability in the D2. Because the $PM_{2.5}$

predictability in South Korea is the focus of this study, most of the experiments were carried out in the D2, while model simulations over the D1 were used to provide the D2 with BCs. Table 3 summarizes the domain descriptions for the WRF and CMAQ model runs.



For another important input field into the CMAQ model simulations, emission data were prepared. KORUS v2.0 emission fields (Jang et al., 2020) were employed for anthropogenic emissions in the two domains. This emission inventory had also

supported official CTM simulations for the KORUS–AQ field campaign in 2016. To prepare biogenic emissions, Model of Emissions of Gases and Aerosols from Nature (MEGAN v2.1; Guenther et al., 2006; Guenther et al., 2012) was run with MODIS land cover data (Friedl et al., 2010), together with MODIS-derived leaf area index (LAI) (Myneni et al., 2002; Yuan et al., 2011). For the MEGAN model runs, the same meteorological fields generated from the WRF model simulations were used. For the considerations of fire emissions, Fire Inventory from NCAR (FINN) was used (Wiedinmyer et al., 2006;

Wiedinmyer et al., 2011).

The observation data used in the EnKF DA experiments were PM$_{2.5}$ data obtained from ground stations located in China and South Korea. We acquired the PM$_{2.5}$ data over China from the China urban air quality real-time data release platform (http://106.37.208.233:20035) managed by the Chinese Ministry of Ecology and Environment, along with another complementary website (http://www.pm25.in). For the PM$_{2.5}$ data over South Korea, the data were downloaded from the

National Ambient air quality Monitoring Information System (NAMIS) of Korea (https://www.airkorea.or.kr). The maximum available observations for PM$_{2.5}$ throughout the period of KORUS–AQ campaign were 866 and 165 in China and South Korea, respectively. Figure 1 shows the locations of those ground stations in D1 and D2.

## 2.4 Experimental Setup

For control run (CTR) without DA, hourly predictions were conducted in D1 by the CMAQ model simulations to generate

the BCs for D2. After that, using the BCs we implemented 24 h CMAQ predictions over D2 each day from 25 April to 12 June, 2016, with the first 5 days for spin-up period, and the 6th day for adapting times for the EnKF DA. To provide the meteorological inputs into the CMAQ model runs over the D2, the WRF model simulations initialized each day 12 h before the CMAQ initialization. In this case, the first 12 h simulations were regarded as spin-up times of the meteorological model. To initialize the next 24 h predictions, the CMAQ model utilized the last hour outputs from the previous 24 h predictions.

The initial ensemble of 40 runs were made, based on the CTR output obtained at 00 UTC on 30 April by perturbing ICs, as described in Sect. 2.1. The ensemble propagations of the CMAQ model simulations started at 00 UTC on 30 April. The DA interval for re-analysis purpose was determined to be 6 h. At the end of the first 6 h prediction (or propagation) of this initial ensemble, the first EnKF DA of PM$_{2.5}$ was conducted at 06 UTC on 30 April, and the updated initial fields from the EnKF DA are termed the 'analysis ensemble' ($\mathbf{x}_{i,k}^{a}$). These analysis states were again propagated until the next EnKF DA step (12

UTC), and were then used as the background state ($\mathbf{x}_{i,k+1}^{f}$) in the next DA step (Eq. 3). Following this process, the analysis–prediction cycle was repeated in the DA sequences to correct the ICs using the EnKF method. Note that the last DA was carried out at 18 UTC on 11 June, and the first three cycles were considered as an adapting time for EnKF. Consequently, the analysis–prediction outputs acquired from the 4 times cycles a day are considered as re-analysis run ('ANL'), rather than predictions. Meanwhile, 24 h predictions (i.e., 24 h DA interval) were also carried out every day, starting from 00 UTC on



01 May to 11 June, with the mean sate of the analysis fields ($\mathbf{x}^a$). A total of 42 day predictions are performed for the prediction run ('PRD') in this study. Figure S1 of the Supplementary Information (SI) shows a schematic of these prediction cycles.

In addition to the CTR run, the two experiments labelled DA_ic (Fig. 2a) and DA_icbc (Fig. 2b) were also made over South Korea (D2). In both the DA_ic and DA_icbc runs, ground-level $PM_{2.5}$ collected in the D2 were assimilated to update the ICs.

The only difference between the two experiments is the process of acquiring the BCs. In the DA_ic experiment, the BCs were obtained from the CTR run over the D1, while in the DA_icbc experiment, the BCs were obtained from the runs with the EnKF DA using ground measured $PM_{2.5}$ collected in China. The technical methods to run the ANL and PRD simulations were the same in both the DA_icbc and DA_ic experiments.

In the DA_ic experiment, we updated only ICs, while in the DA_icbc experiment, we updated both the ICs and the BCs. The

goals of this experimental setup are to make it possible to evaluate how much and to what degree the EnKF DA technique could enhance the $PM_{2.5}$ prediction skills, and to separately estimate the contributions of the improved ICs and BCs to the predictabilities of $PM_{2.5}$ over South Korea.

## 3 Results and Discussion

### 3.1 Impact of the improved initial fields

Figure 3 shows the daily variations of surface $PM_{2.5}$ from 1 May to 12 June, 2016 (KORUS–AQ period). In Fig. 3, the observations (OBS), denoted by open circles, were obtained by averaging all the ground $PM_{2.5}$ available in South Korea (we call this the "aggregation plot"). The simulation results (CTR, ANL with the 3DVAR, and ANL with the EnKF) were also calculated by averaging the model outputs at the corresponding observation locations. Figure 3 shows that the control run (CTR) without DA (solid blue line) tended to consistently underestimate the daily averaged $PM_{2.5}$ throughout the simulation

period. The ANL simulation with the EnKF (solid red line) showed the best agreement with the observations. The performances of the EnKF are also found to be better than those of the 3DVAR (dashed purple line).

Figures 4(a) and (b) present the horizontal distributions of surface $PM_{2.5}$ for background and analysis fields at a specific EnKF DA sequence, respectively. Here, the "analysis field" indicates the initial concentration fields updated by the EnKF method. Figures 4(a) and (b) also plot the observed $PM_{2.5}$ used in the DA with the same color-scale. Figure 4(c) gives the

analysis increments representing the extent of the corrections of $PM_{2.5}$ by the EnKF data assimilation. Again, the background fields tended to underestimate the $PM_{2.5}$ over the inland areas. As discussed in Sect. 2.4, Fig. 4(b) was obtained using an average of 40 analysis ensembles. It can be seen how the estimated background error covariance with a short-term ensemble propagation could correct the model background by assimilating observations. In a relatively isolated ground station, such as Jeju Island (the location is shown in Fig. 1), the analysis increments occurred largely in the down-wind area (Fig. 4c). This

provides a clear example of flow-dependent correction of the EnKF technique.





Figure 5 presents the average diurnal variations generated by aggregating the PM$_{2.5}$ data during 42 days from all the observation sites. The vertical bars indicate one standard deviation of the averaged samples. Figure 5 shows a clear pattern of the results from each simulation showing distinct diurnal variations. This pattern appears to be caused mainly by the changes in meteorological fields during the day. During daytime, relatively high mixing height due to the thermal and mechanical

development of boundary layers could lead to decreased PM$_{2.5}$ within the boundary layers. In contrast, after sunset, PM$_{2.5}$ started to increase, because the mixing height became shallow, due to the stable atmospheric conditions caused by sunset, as well as the weak wind speeds. This diurnal pattern was also found in the observation data, but their variations are weaker than those from the model simulations. The CTR experiment again consistently underestimated the diurnal PM$_{2.5}$ throughout a day. However, quite good agreement with the observed PM$_{2.5}$ was found in the ANL simulations with the EnKF (solid red

line). Focusing on the mean values only at each DA time (00, 06, 12, and 18 UTC), the updated concentrations for the 3DVAR simulations (purple triangles) are always closer to the observations than those for the EnKF simulations (red triangles). This indicates that the 3DVAR used larger model errors with higher uncertainties than those of the EnKF, when the DA process was carried out. However, the EnKF showed better performance than the 3DVAR simulations in the following time, especially during the daytime (e.g., 01 UTC to 06 UTC, and 06 UTC to 12 UTC), although its correction

strength by assimilation is lower than the 3DVAR. We believe this is because the flow-dependent characteristics of model errors in the EnKF technique improve the model fields more realistically than those in the 3DVAR. In contrast, the 3DVAR uses a "static" climatological BEC, which usually represents a semi-Gaussian distribution. The better results from the EnKF method (than the 3DVAR method) can also be attributed to the realistic considerations of vertical mixing within the boundary layer in the BEC (Pagowski and Grell, 2012). More sophisticated comparisons in the configurations, such as error

variances, observation operator, and vertical length scale of the BEC, are necessary in future study for a direct comparison of the two DA algorithms. To more quantitatively evaluate the performances of the 3DVAR and the EnKF techniques, Table 4 also summarizes the statistical metrics based on the reanalysis outputs (ANL). Section 3.4 below discusses the quantitative evaluation in more detail.

## 3.2 Impact of the improved boundary conditions

In the previous section, we examined the effects of the initial fields (the DA_ic experiment) in South Korea. The influences of the updated ICs tend to quickly disappear with time over the relatively small domain (D2), particularly when atmospheric flows are fast. In this section, we conducted additional assimilation with the ground observations from China in the D1, in addition to the data assimilation with ground observations from South Korea (the DA_icbc experiment). The DA_ic and DA_icbc experimental results were again compared in South Korea, which is our main domain of interest. Although the

prediction strategy (refer Fig. S1 of the SI) was the same in the DA_icbc experiment, only PRD runs are shown in this section for simplicity.

Figure 6 shows the averaged PM$_{2.5}$ used for the four lateral boundaries of the domain 2 in both the DA_ic and DA_icbc experiments. At the four lateral boundaries, PM$_{2.5}$ was averaged over 6 weeks, and Fig. S2 of the SI shows the south, east,





north, and west boundaries of domain 2 (D2). The color-filled contours on the vertical planes correspond to longitudinal

direction from west to east (latitudinal direction from south to north), and vertical direction for the southern and northern (western and eastern) boundaries of domain 2. Together with the $PM_{2.5}$, Fig. 6 also plots the mean wind velocity across the four boundaries, to show the inflow into the D2 and outflow from the D2 with positive (solid) and negative (dashed) contour lines, respectively. In the upper panels of Fig. 6, western part of the north boundary and northern part of the west boundary show relatively high $PM_{2.5}$ ($> 15 \, \mu g \, m^{-3}$) within 1 km altitude. Although long-range transport of air pollutants from China to

South Korea sometimes occurs in the upper layers, the averaged $PM_{2.5}$ at the northwest boundaries were high within the boundary layer. We should also note in Fig. 6 that the northwestern boundary had a strong inflow that could result in high $PM_{2.5}$ in the D2.

The middle and bottom panels of Fig. 6 show that at all the boundaries, the DA_icbc experiment exhibited higher $PM_{2.5}$ than the DA_ic experiment. This indicates that the control simulation without assimilation (CTR) over the D1 under-calculated

$PM_{2.5}$ in China. Figure 6c shows that there are small changes in $PM_{2.5}$ above 2 km altitude, while the changes become larger within the boundary layers. To quantify the amounts transported into and out of the D2, we calculated the $PM_{2.5}$ fluxes by multiplying $PM_{2.5}$ by wind velocities, and then averaged them over the simulation period (refer Fig. S3 of the SI). The cross-sectionally averaged $PM_{2.5}$ flux at the west boundary increased from 19.2 to 26.6 $\mu g \, m^{-2} \, s^{-1}$ from the DA_ic to DA_icbc experiments. This indicates that larger amounts of $PM_{2.5}$ were actually transported long-distance from China to South Korea,

mainly through the northwestern boundary of domain 2 during the KORUS–AQ period.

A ground station where the influences of the boundary conditions can be checked is Baekryeong-do, South Korea (shown with star symbol in Fig. 1). The reason is that Baekryeong-do is located at the west-end of domain 2 (very near the western boundary of the D2), and is also minimally affected by local inland emissions (i.e., there are no major industries, and only a small population living on the island). Figure 7a shows the averaged diurnal variations of $PM_{2.5}$ at Baekryeong-do evaluated

from D1. Hence, the results with (solid red line with triangles) and without (blue dashed line with rectangles) the DA can be perceived as the boundary conditions in D2 for the DA_icbc and DA_ic experiments (refer Fig. 2), respectively. The averaged diurnal variation of $PM_{2.5}$ without the DA is in the range between 10 and 20 $\mu g \, m^{-3}$, which is approximately 10 $\mu g \, m^{-3}$ lower than the observed $PM_{2.5}$. However, when the assimilation of the observations in China was applied, almost the same levels of $PM_{2.5}$ as the observations were reproduced. We found that the 24 hour predictions that were evaluated at the

same location in D2 were greatly improved (Fig. 7b). This is confirmed by the results from the DA_icbc experiment in Fig. 7b. Since the observed $PM_{2.5}$ at the Baekryeong-do site were assimilated to improve the initial conditions in both the DA_ic and DA_icbc experiments, the predictions started from the similar $PM_{2.5}$ to the observed $PM_{2.5}$. However, the predictions for the DA_icbc experiment agreed greatly with the observed $PM_{2.5}$ due to the application of accurate boundary conditions, while the prediction for the DA_ic experiment rapidly converged to the CTR run, because of the same boundary conditions

as the CTR run. Another fact we should note is that analysis increments by assimilating Baekryeong-do data for the DA_icbc experiment in D2 would be minimal, as the background $PM_{2.5}$ was already close to the observed $PM_{2.5}$ because of the updated boundary conditions.





Figure 8 presents the daily variations of $PM_{2.5}$ like in Fig. 3, except for the results from the "PRD runs" for the DA_ic and DA_icbc experiments. The PRD runs are technically the same as the ANL runs except for the prediction lead time (of 24 and

6 hour, respectively). Again, significant improvements in the DA_ic and DA_icbc experiments were found, compared to the results from the CTR run. When the dominant synoptic wind directions were southerly or easterly (e.g., on June 2 to 4), there were only small differences between the DA_icbc and DA_ic experiment, and thus limited improvements were achieved. Similarly, no improvements for updating the boundary condition in the DA_icbc experiment were found during the precipitation days on May 10 and 24, and on June 1 and 6. However, large improvements could be made, when the Yellow

dust event occurred during May 4 to 7, and when the westerly winds prevailed over the D2 between May 20 and 27 (except on 24 May). Therefore, to improve $PM_{2.5}$ predictability in South Korea, it is of great importance to provide the appropriate boundary conditions by assimilating the ground observation data in the upwind area (i.e., EC, NC, and NEC region, refer Fig. 1).

To evaluate the $PM_{2.5}$ predictability in South Korea, Fig. 9 also displays the averaged diurnal variations, and compares the

prediction runs (PRD) for the two experiments, DA_ic and DA_icbc. The performances of 24 hour predictions were launched every 00 UTC. The predicted $PM_{2.5}$ for the PRD runs show better performances than the CTR run with reduced errors and biases, although the biases are larger than those for the ANL runs (shown in Fig. 3). Again, the averaged diurnal $PM_{2.5}$ for the DA_icbc experiment is closer to the observations than that for the DA_ic experiment. This is because the enhanced boundary information was repeatedly provided at the everyday prediction sequence. Also, the negative biases

found in the DA_ic experiment were greatly alleviated with the DA_icbc experiment, even if the same emissions and meteorological fields were applied for the 24 hour predictions. The immediate improvements could be seen immediately after the predictions started at 00 UTC. As time progressed, the biases and errors were also propagated. However, the biases in the DA_icbc experiment became about half of those in the DA_ic experiment. Note also that the slightly over-predicted $PM_{2.5}$ for the DA_icbc experiment between 18 and 23 UTC were caused by insufficient information about vertical mixing

during night-time. Simulated nocturnal boundary layer heights were lower than real boundary layer heights. This is a critical problem in meteorological modeling, and has been discussed in many previous publications (Eder et al., 2006; Hong, 2010).

### 3.3 Statistical Evaluations: Quantification of contributions by updating the initial and boundary conditions

Table 4 summarizes the statistical performance metrics that we calculated to evaluate the model performances. Table S1 of the SI gives the mathematical definitions for the performance metrics. The average $PM_{2.5}$ over the entire simulation period

was 27.9 μg m$^{-3}$, and the CTR run produced the under-estimated $PM_{2.5}$ of 17.9 μg m$^{-3}$. The application of the updated initial conditions (the DA_ic experiment) improved the average $PM_{2.5}$, which was 25.4 and 22.9 μg m$^{-3}$ for the ANL and PRD runs, respectively. The results for the DA_icbc experiment were even closer to the observations than those of the DA_ic. The average $PM_{2.5}$ from the ANL and PRD runs for the DA_icbc experiment was 26.5 and 25.5 μg m$^{-3}$, respectively. All the statistical metrics for the DA_icbc experiment were improved, compared to the DA_ic experiment. Focusing on the

PRD runs, the IOA increased significantly from 0.610 to 0.665 (for the DA_ic experiment), and then to 0.685 (for the





DA_icbc experiment). The average RMSE was reduced from 20.8 to 18.3 µg m$^{-3}$ for the DA_icbc experiment, while it was 18.8 µg m$^{-3}$ for the DA_ic experiment. This small reduction in the averaged RMSE between the DA_ic and the DA_icbc experiments can be attributed to the over-prediction of PM$_{2.5}$ during the night-time, as Fig. 9 shows. On the other hand, remarkable improvements were found in the MBs. In the DA_ic experiment, the averaged MB was drastically reduced from

-10.2 to -5.3 µg m$^{-3}$. Another large reduction in the averaged MB was found from -5.3 to -2.5 µg m$^{-3}$ from the DA_ic experiment to the DA_icbc experiment. The normalized MB (NMB) was also reduced by 17.3 % in the DA_ic experiment from 36.2 % for the CTR run. In addition, the DA_icbc experiment led to another considerable decrease in the NMB by 9.0 %, compared to the DA_ic experiment.

To investigate the quantitative contributions of the initial and boundary conditions to the model performances, we calculated

the 'rate of improvement (ROI)' with respect to the PRD results (see Table 5). The ROIs are defined by the ratios of enhanced (R and IOA) or reduced (RMSE and MB) amounts of the corresponding statistic metrics to those calculated from the CTR run. Based on the ROI for the DA_ic and DA_icbc experiments, we can estimate the ROIs associated with the initial correction (the DA_ic) and the boundary correction (the DA_bc). The ROIs for the DA_ic and DA_icbc experiments were 10.2 and 15.0 % in terms of R (Pearson's correlation coefficient), respectively. Therefore, the estimated ROI due to the

DA_bc might be 4.8 %. The contributions in MB can also be estimated quantitatively in terms of the ROIs. Updated boundary conditions resulted in an improvement of 27 % in MB in terms of ROI. In the case of the applications of the DA_ic and the DA_bc, the ROIs were 9.0 and 3.3 % increase in terms of IOA, and 9.6 and 2.4 % decrease in terms of RMSE, respectively.

## 4 Conclusions

To improve PM$_{2.5}$ prediction in South Korea, we developed and applied an EnKF data assimilation method to the WRF–CMAQ modeling system. For the data assimilation, we employed two groups of ground observations from China and South Korea. We found that when we updated the ICs via the EnKF data assimilation, the PM$_{2.5}$ predictions in South Korea could be greatly improved. In comparative analysis between EnKF and 3DVAR, the EnKF technique showed better performance than the 3DVAR in short-term PM$_{2.5}$ predictions. These results indicate that the BEC used in this study can realistically

reflect current states of the atmosphere, particularly in the boundary layer.

This study also highlighted the importance of updating boundary conditions to further enhance the PM$_{2.5}$ predictability over South Korea. Long-range transport from China directly impacts the air quality in South Korea, particularly during high PM$_{2.5}$ episodes. Since there are only restrictive effects of the DA with ground observations inside South Korea on improvement in analysis fields and predictions, we updated the inflow boundary conditions via the EnKF DA that uses the observations in

China. Comparison of the studies with and without the updated BCs suggested that improved initial conditions (the DA_ic experiment) reduced the NMBs from -36.2 to -18.9 % compared to the control run, and even further updating the initial and boundary conditions (the DA_icbc experiment) improved the NMBs from -36.2 to -9.9 % in terms of the 24 hour PM$_{2.5}$





prediction over South Korea. In terms of IOA (in terms of MB), the contributions of updating the ICs and BCs to 24 hour predictability were estimated to be 73 and 27 % (63 and 37 %), respectively. However, caution should be exercised, in that these estimations are made only for a specific period (KORUS–AQ campaign), and can vary with atmospheric conditions. A longer period test is needed for general quantification.

Recently, the EnKF has also been used to assimilate satellite-retrieved aerosol observations (e.g., Sekiyama et al., 2010; Schutgens et al., 2010a and 2010b; and Yin et al., 2016). Other groups also used the EnKF method for a joint optimization of initial conditions and emission scaling factor (e.g., Tang et al., 2011; Peng et al., 2017 and 2018). Given that we have shown that the consideration of the transboundary air pollution is of significance in the PM$_{2.5}$ predictions over South Korea, assimilating aerosol optical depth (AOD) data from the satellites over the Yellow Sea (where no ground observations are available) is expected to provide the PM$_{2.5}$ prediction system with important information.

Throughout this study, the DA method of 'perturbed observation EnKF' (first proposed by Evensen, 2003) was employed. However, there are some popular variants of the EnKF method that obviate the need to perturb observations, such as Ensemble Square Root Filter (EnSRF; Whitaker and Hamill, 2002), Ensemble Adjustment Kalman Filter (EAKF; Anderson, 2001), and Local Ensemble Transform Kalman Filter (LETKF; Hunt et al., 2007). Two of these EnKF variants are also being tested to alleviate the sampling errors in the observation ensemble, and the results will also be reported in the near future in the context of further development of the ensemble data assimilations and the Korean air quality prediction system.

**Acknowledgment**

This research was supported by the FRIEND (Fine Particle Research Initiative in East Asia Considering National Differences) Project (2020M3G1A1114617) and Basic Science Research Program (2021R1A2C1006660) of the National Research Foundation of Korea (NRF) grant funded by the Ministry of Science and ICT (MSIT).

**Code and data availability**

The WRF model v3.8.1 (doi:10.5065/D6MK6B4K) is available after user registration through the web page (https://www2.mmm.ucar.edu/wrf/users/download/get_source.html). The CMAQ model v5.1 (doi:10.5281/zenodo.1079909) is open-source and can be downloaded at https://github.com/USEPA/CMAQ. The EnKF method and related processes written in IDL language are available at https://doi.org/10.5281/zenodo.5376214 (Park, 2021). We uploaded the model outputs for the ensemble mean with the netCDF binary format and all the assimilated observation data at https://doi.org/10.5281/zenodo.5566441/.





## Author contributions

SYP and CHS designed this study and experiments. SYP, UKD, KY, and IU developed the EnKF code and discussed the results. SYP and UKD carried out the simulations, produced the figures, and prepared the initial manuscript draft. JY performed the 3DVAR experiments and provided all the input data for the CMAQ model. CHS contributed to the final writing with the comments from all co-authors.

## Competing interests

The authors declare that they have no conflict of interest.

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



**Table 1. WRF model configurations selected in this study.**

| Parametrization | WRF option |
|---|---|
| Planetary boundary layer | Yonsei University (YSU) scheme (Hong et al., 2006) |
| Microphysics | WRF single-moment 6-class (WSM6) scheme (Hong and Lim, 2006) |
| Cumulus parameterization | Grell–Freitas ensemble scheme (Grell and Freitas, 2014) |
| Land surface model | Noah–MP (Niu et al., 2011; Yang et al., 2011) |
| Shortwave/longwave options | Rapid Radiative Transfer Model for Global Circulation Models (RRTMG) (Iacono et al., 2008) |
| Surface layer options | revised MM5 scheme for Jiménez et al. (2012) |


**Table 2. CMAQ model configurations selected in this study.**

| Parametrization | CMAQ option |
|---|---|
| Aerosol thermodynamics | AERO6 (Appel et al., 2013) |
| Gas-phase Chemistry | SAPRC07tc (Hutzell et al., 2012) |
| Chemistry solver | Euler Backward Iterative (EBI) chemistry solver (Hertel et al., 1993) |
| Dry deposition | M3DRY (Pleim and Xiu, 2003) |
| Horizontal advection | Yamo global mass-conserving scheme (Yamartino, 1993) |
| Vertical advection | Vwrf-Piecewise Parabolic Method (Colella and Woodward, 1984) |
| Horizontal diffusion | Multiscale (Louis, 1979) |
| Vertical diffusion | Asymmetric Convective Model, version 2 (ACM2; Pleim, 2007a, b) |

**Table 3. Domain descriptions for WRF and CMAQ models.**

| Model | WRF v3.8.1 | | CMAQ v5.1 | |
|---|---|---|---|---|
| Domain | D1 | D2 | D1 | D2 |
| Horizontal grids | 153×114 | 109×109 | 144×105 | 100×100 |
| Grid resolution | 27 km | 9 km | 27 km | 9 km |
| Vertical layers | 33 layers (top: 50 hPa) | | 15 layers (top: 20 km) | |
| ICs and BCs | NCEP FNL 1° data | | Predefined clean profiles | |
| Target periods | 00 UTC 01 May 2016 – 00 UTC 12 June 2016 | | | |




**Table 4. Statistical metrics for the experiments of DA_ic and DA_icbc. Experiments were evaluated for the 6 hourly assimilated analysis run (ANL), and for the one-day prediction run (PRD). The ANL run using 3DVAR in the DA_ic experiment is included for comparison.**

| Experiments and simulations | | MEAN* ($\mu g\ m^{-3}$) | R | IOA | RMSE ($\mu g\ m^{-3}$) | MB ($\mu g\ m^{-3}$) | NMB (%) |
|---|---|---|---|---|---|---|---|
| **CTR** | | 17.9 | 0.421 | 0.610 | 20.8 | -10.2 | -36.2 |
| **DA_ic** | **ANL (3DVAR)** | 22.1 | 0.618 | 0.761 | 15.6 | -5.8 | -20.8 |
| | **ANL** | 25.4 | 0.646 | 0.795 | 14.3 | -2.5 | -9.0 |
| | **PRD** | 22.9 | 0.464 | 0.665 | 18.8 | -5.3 | -18.9 |
| **DA_icbc** | **ANL** | 26.5 | 0.656 | 0.804 | 14.1 | -1.4 | -5.1 |
| | **PRD** | 25.5 | 0.484 | 0.685 | 18.3 | -2.5 | -9.9 |

\* Mean concentration in observed data is **27.9 $\mu g\ m^{-3}$**.


**Table 5. Rate of improvement (ROI) by EnKF data assimilation in one-day predictions. The ROI is the ratio of the enhanced (R and IOA) or reduced (RMSE and MB) statistical metrics to those for CTR simulation. The ROI by the updating boundary conditions (DA_bc) can be estimated from the difference between that obtained by the DA_ic and DA_icbc experiments.**

| | DA_ic | DA_icbc | Estimated DA_bc |
|---|---|---|---|
| **R** | 10.2 % | 15.0 % | 4.8 % |
| **IOA** | 9.0 % | 12.3 % | 3.3 % |
| **RMSE** | 9.6 % | 12.0 % | 2.4 % |
| **MB** | 48 % | 75 % | 27 % |




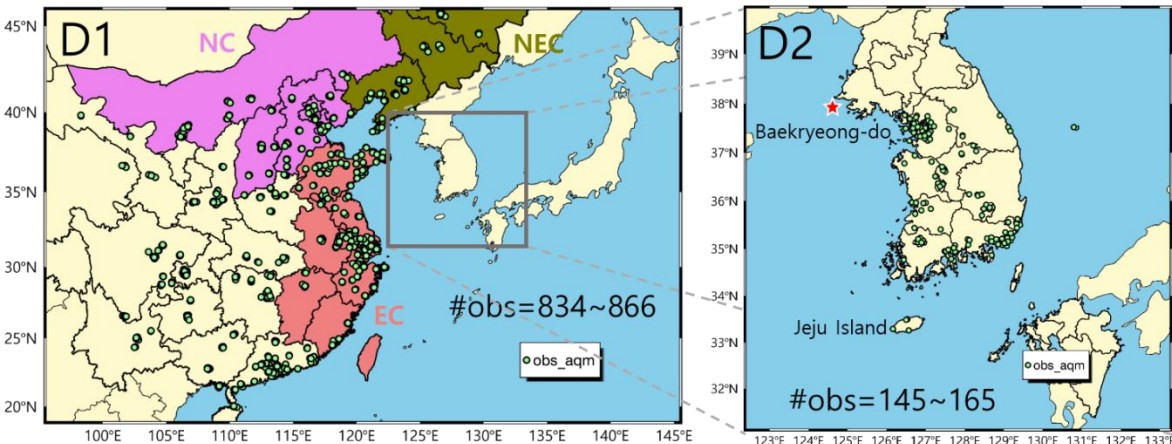

**Figure 1. Simulation domains with nested modeling. D1 and D2 represent mother and daughter domain, respectively. The locations of ground stations in China (D1) and South Korea (D2) are marked on the maps with green dots. In the D1 (left), Northeast China (NEC), North China (NC), and East China (EC) regions that frequently influence air quality in South Korea are grouped with olive, violet, and coral colors, respectively. The star symbol with red color indicates Baekryeong-do observatory, where the evaluation of boundary inflow was made. Jeju Island in D2 (right) is an ideal location to see the flow-dependent correction by the EnKF DA. The total number of available stations used in EnKF data assimilation is also shown in both domains.**






**(a) DA_ic**

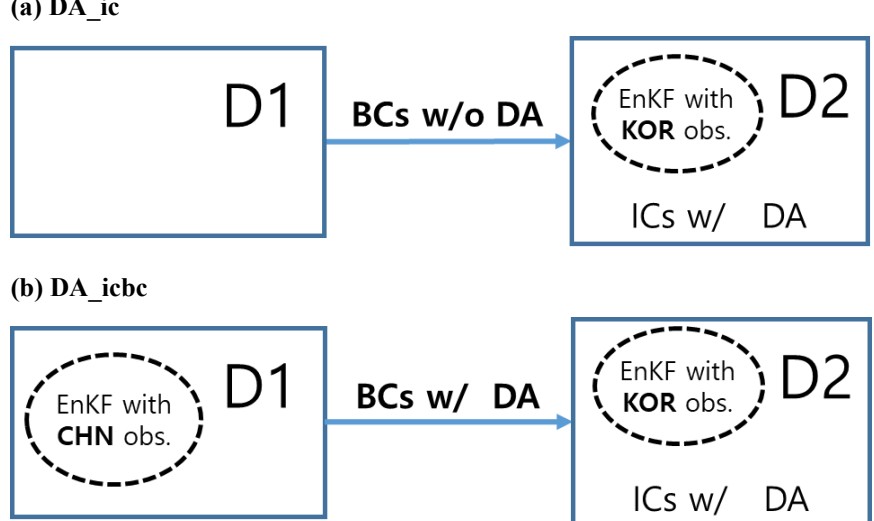

**(b) DA_icbc**

**Figure 2. Schematic flow-chart for the experiments performed in this study. To evaluate PM$_{2.5}$ predictability in South Korea, (a) DA_ic experiment updates the initial conditions (ICs) only within D2, while (b) DA_icbc experiment provides D2 with updated boundary conditions (BCs) via assimilating ground observations in China (CHN obs.), and also updating the I.C.s for D2 using Korean ground observations (KOR obs.).**





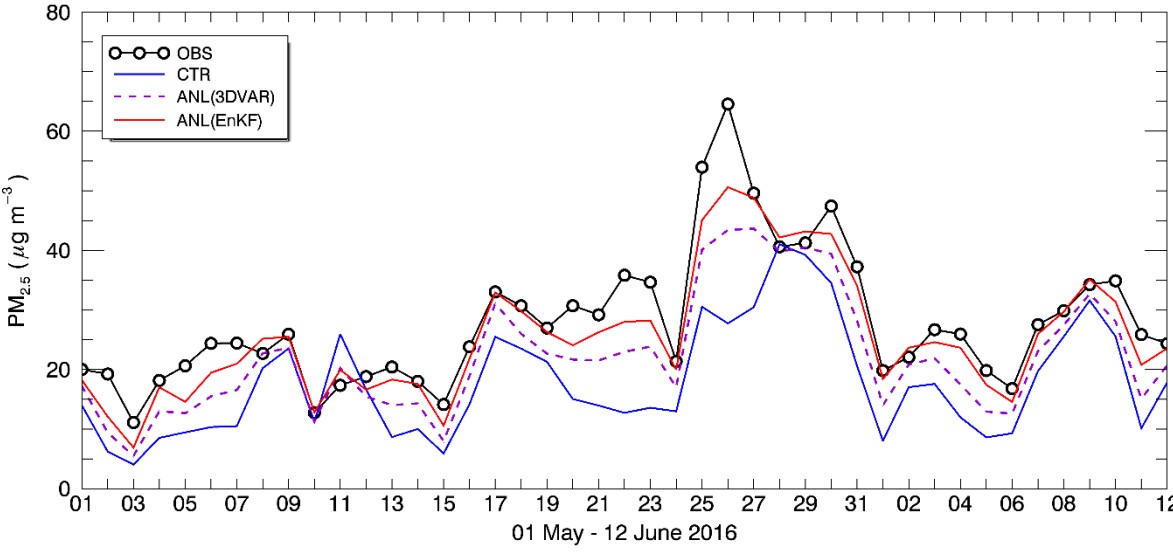

**Figure 3. Daily variations of surface PM$_{2.5}$ for DA_ic experiment. Observations (OBS) are represented by black solid line with open circles. Model results for control (CTR) run without DA, for reanalysis (ANL) run with 3DVAR, and reanalysis ANL with EnKF are plotted by blue solid line, purple dashed line, and red solid line, respectively. Values were prepared from daily averages at all the observation sites in South Korea (D2).**





(a) Background       (b) Analysis       (C) Analysis increments

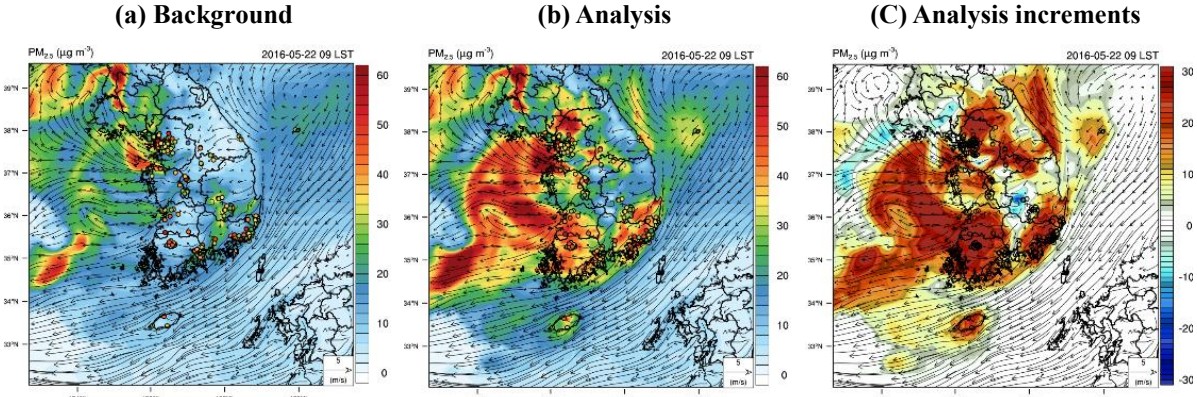

**Figure 4. Snap shots of the horizontal distributions of PM$_{2.5}$ before ((a) background) and after ((b) analysis) the application of EnKF technique at 00 UTC on 22 May, 2016. The observed concentrations are also shown on the map with the same color-scales as contour values. In the right-hand panel (c), the analysis increments are also presented, and flow-dependent corrections can be visible when the wind vectors are overlaid with the analysis increments. The big island in the Southern Sea of the Korean Peninsula is Jeju Island, where the flow-dependent behavior can be noticed.**




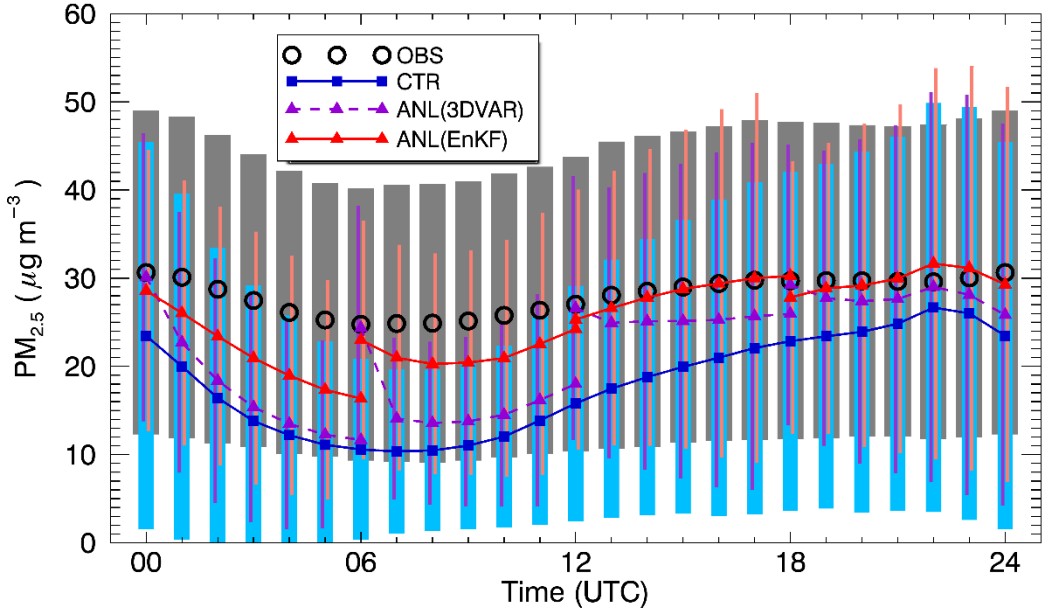

**Figure 5.** Average diurnal variations of PM$_{2.5}$ aggregated from all ground stations in South Korea (D2) for the DA_ic experiment. The color labels are the same as in Fig. 1, except for symbols. The error bars with gray, cyan, purple, and pink indicate one standard deviation ($\pm\sigma$) for OBS, CTR, ANL by 3DVAR, and ANL by EnKF simulations, respectively.




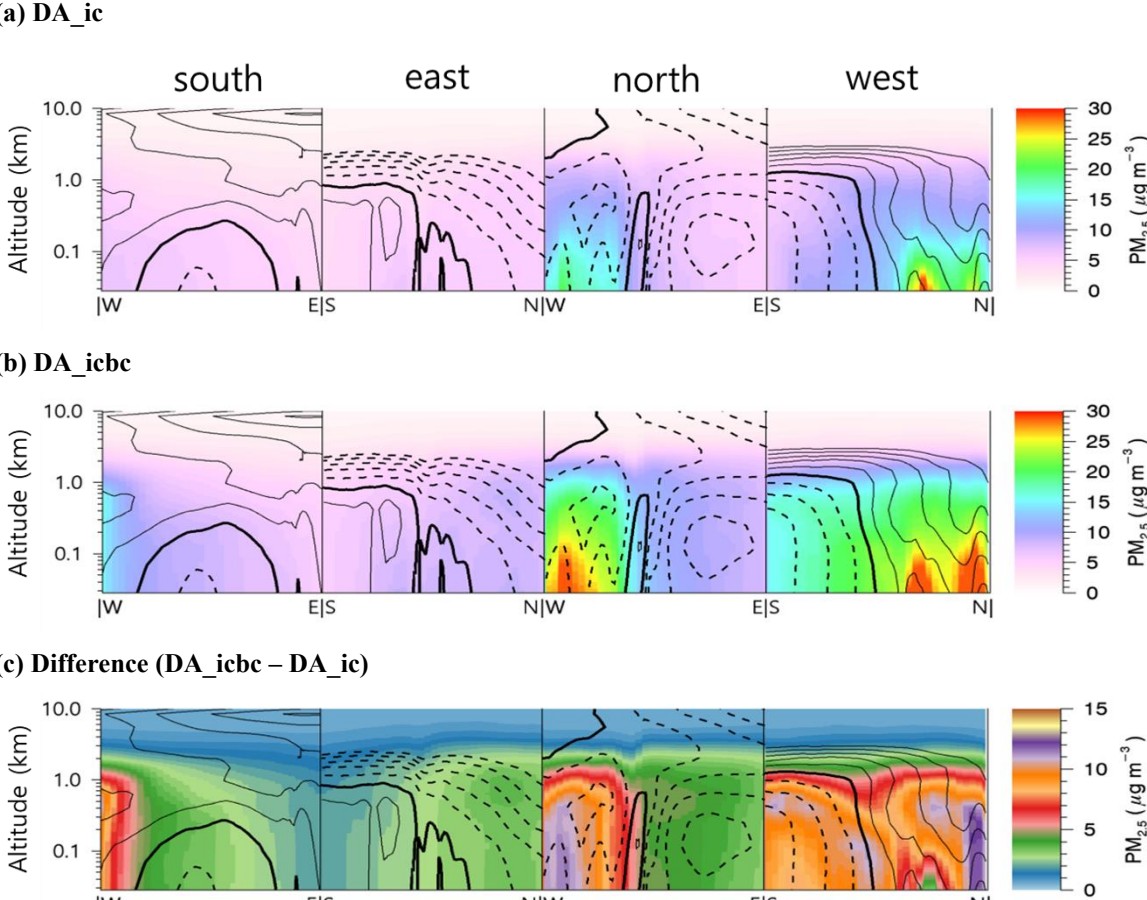

**Figure 6. Averaged PM$_{2.5}$ distributions in the four lateral boundaries of the domain 2 (D2: south, east, north, and west from the left to the right, refer Fig. S3 of the SI). Each panel includes black contour lines that explain the inflow (solid lines) and outflow (dashed lines) wind vector with 1 ms$^{-1}$ interval. The thick black lines indicate zero wind speed. In (a) DA_ic and (b) DA_icbc, the averaged lateral boundary concentrations are provided into D2 without and with the EnKF data assimilation in China (D1), respectively. The increments in DA_icbc experiment are also presented in the bottom panel (c). Note that the y-axis for altitude is presented in log-scale, to emphasize the results below the boundary layer.**



**(a) Baekryeong-do site in D1**

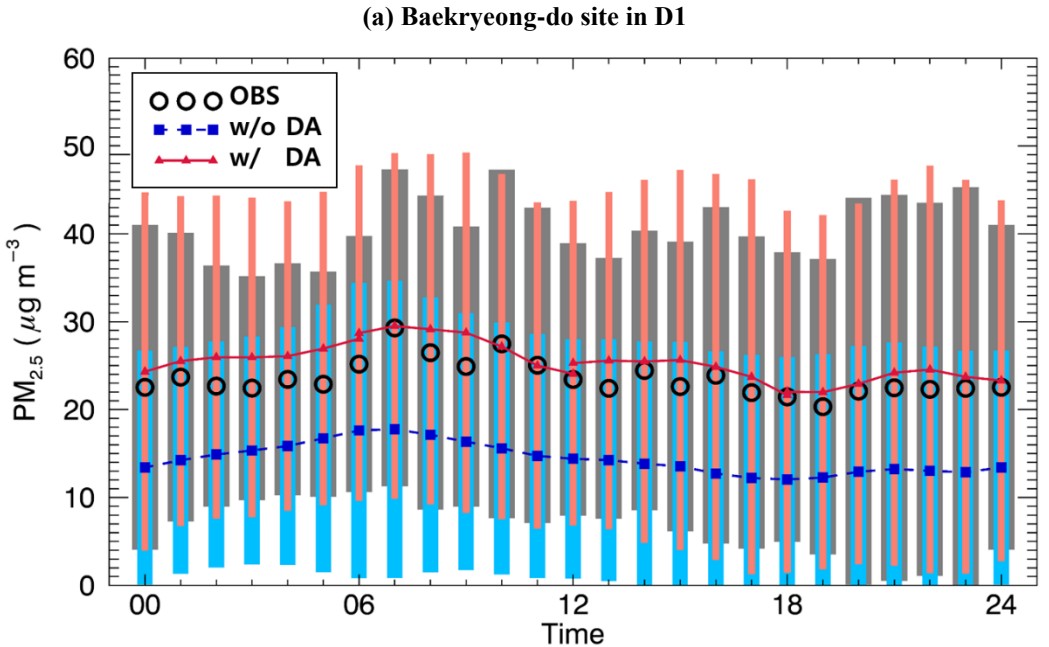

**(b) Baekryeong-do site in D2**

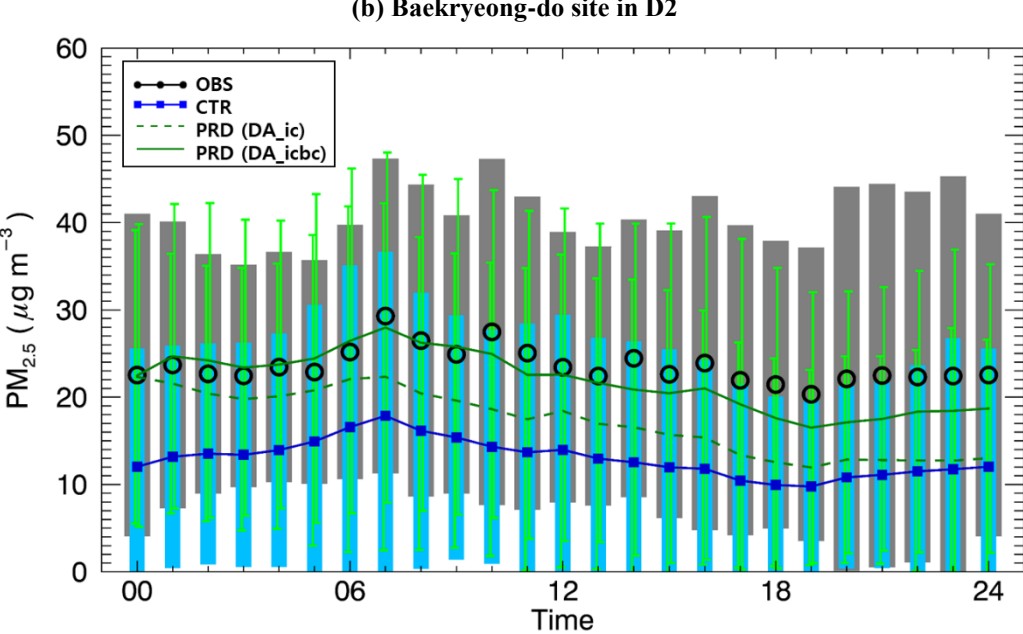

**Figure 7. Averaged diurnal variations of PM$_{2.5}$ aggregated at Baekryeong-do site from the results obtained in (a) D1, and (b) D2. The line colors and symbols are the same as in Fig. 5, except for the prediction runs in D2, which are plotted by dashed and solid green lines for DA_ic and DA_icbc experiments, respectively, in panel (b).**



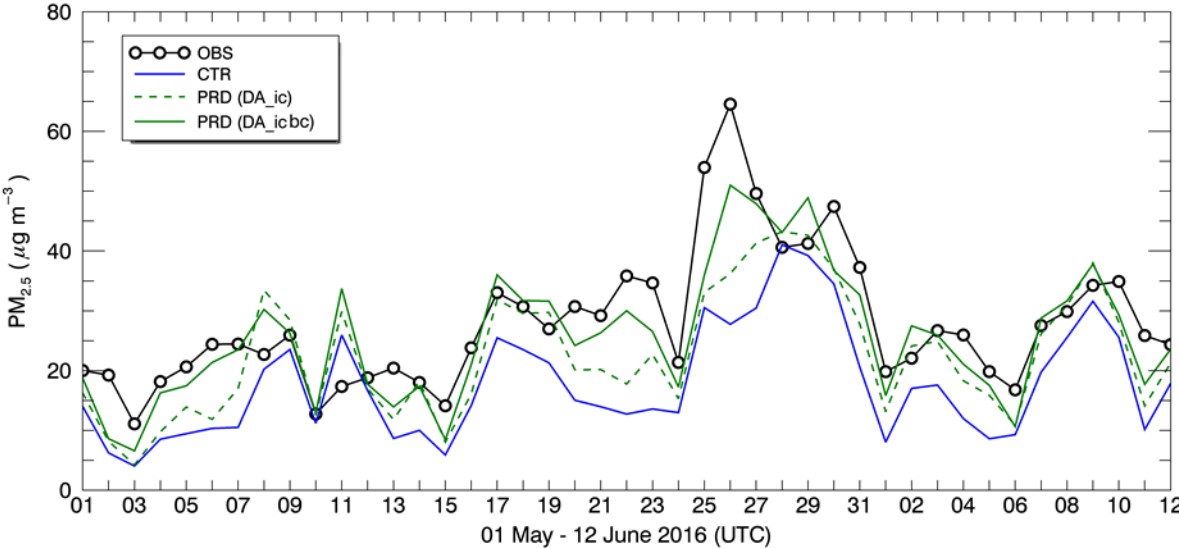

**Figure 8. Daily averaged variations of PM2.5. The lines and colors are the same as in Fig. 3, except for the one-day prediction runs (PRD). One-day predictions only with updated initial condition (DA_ic), and with initial and boundary conditions (DA_icbc), are presented by the dashed and solid green lines, respectively.**




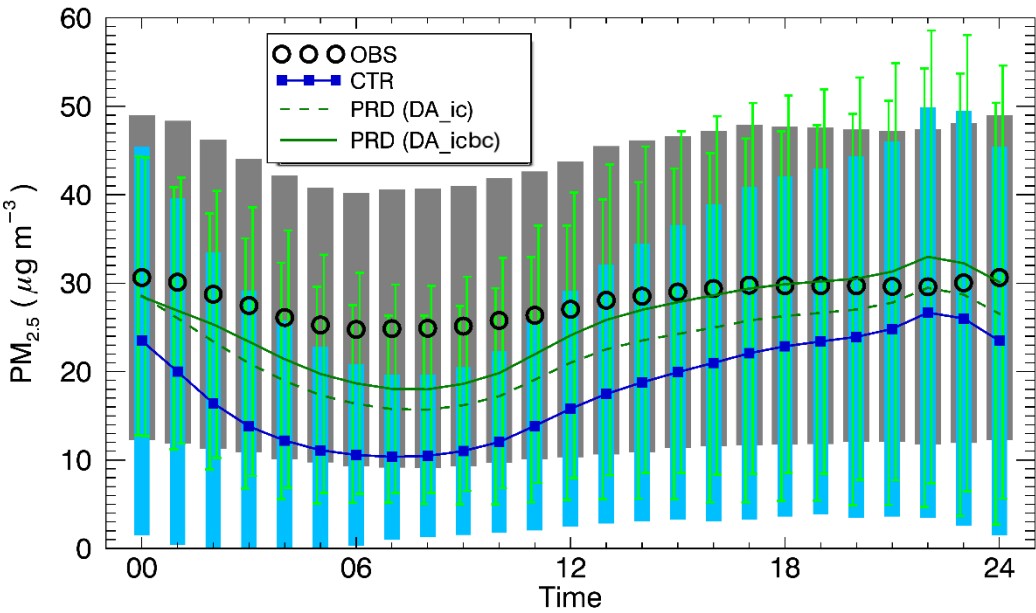

**Figure 9. Averaged diurnal variations of PM$_{2.5}$ aggregated from all ground stations in South Korea (D2). The color and symbols are the same for observations (OBS) and control run (CTR) as in Fig. 5. One-day predictions only with updated initial condition**
**(DA_ic), and with initial and boundary conditions (DA_icbc), are presented by dashed and solid green lines, respectively. One standard deviation (σ) is also plotted for each case using vertical bars. The left and right vertical bars indicate ±σ for DA_ic and DA_icbc, respectively.**