# Peer review of "Implementation of an Ensemble Kalman Filter in the Community Multiscale Air Quality Model (CMAQ Model v5.1) for Data Assimilation of Ground-level $PM_{2.5}$"

_Geoscientific Model Development, 2021_

## Author Comment (AC1)

**Response to Reviewer 1:**

First of all, thank you for your constructive comments on this manuscript. We have tried to address all the comments you made in the manuscript and also in the replies. The line numbers mentioned in the following replies are based on the revised manuscript with track changes.

■ General

This manuscript presented method comparison: EnKF and 3D-Var, for assimilating surface PM2.5 observations with two settings: IC and ICBC. It is a straightforward paper. One major issue is that the prediction model CMAQ has only 15 layers up to 20km, which is too coarse. How many layers below 1km? Could this coarse vertical resolution cause artificial dilution for near-surface pollutant concentration, and result in the systematic PM2.5 underestimation? Although this manuscript focuses on data assimilation (DA), the corresponding prediction model should be reasonable, too. Otherwise, the DA methods only show their effect on correcting the systematic underestimation.

**Reply:** Because surface $PM_{2.5}$ is major focus of this study, a total of eight layers out of 15 layers are located below 1 km. The mid-point height of the first layer is averagely 16 m above the ground level, which means that the first layer depth is averagely 32 m. We reasonably assumed in this study that all the observation instruments are installed within the first layer. In addition, we believed that 32 m depth of the first layer and 8 layers below 1 km may be sufficient to avoid artificial dilution. We actually conducted a simple sensitivity test with varying vertical resolutions to investigate the impacts of the number of the vertical layers on the ground-level $PM_{2.5}$ in the prediction models such as CMAQ. In Fig. R1, the simple test shows almost no impacts made by doubling the number of the vertical layers (i.e. nz=15 vs. nz=30). Based on this sensitivity test, we decided not to increase the vertical resolution in our simulations because of severe increases in the computational costs, especially for this type of ensemble approach. Instead of adding these sensitivity results into the manuscript or supplementary information, we have decided to add information on the corresponding altitude for the vertical layers beneath Table 3 (please see, **line 773**).

As you mentioned in your comments, surface $PM_{2.5}$ has been underpredicted by many prediction models in East Asia, which is mainly due to the underpredictions of the concentrations of SOA (Secondary Organic Aerosols) and fugitive dust (Volkamer et al., 2006; Philip et al., 2017; Jeong and Park 2019). This is a well-recognized (and also well-defined) problem. We have therefore attempted to correct these underpredictions of both SOAs and fugitive dust. However, we believe that correcting the underpredictions of $PM_{2.5}$ is a different story from this work. Thus, we have been omitting this issue/topic in this manuscript.

[Figure]

Figure R1. Sensitivity tests of the vertical resolution to ground PM$_{2.5}$ simulations.

■ Comments

**1. Section 2. PM2.5 is not a single species in CMAQ. How do you map the PM2.5 increment to individual CMAQ aerosol species?**

**Reply:** Because PM$_{2.5}$ was only control variable, we applied the increment ratio (i.e. analysis over background, $\mathbf{x}^a/\mathbf{x}^f$) to all the aerosol species, following the original contributions to PM$_{2.5}$. In this case, we don't need to consider many aerosol species in the observation operator to calculate PM$_{2.5}$. We used a post-processing tool included in the CMAQ software to calculate PM$_{2.5}$ before the DA process. For the sake of the readers' understanding, we have added some sentences explaining how we updated the CMAQ aerosol species (please, refer to **lines 143–147**).

**2. Page 9, line 155. What's the vertical extent of the 50% perturbation being applied, to all layers? Considering that it is used to the assimilating surface observation, certain justification is needed.**

**Reply:** Yes, we perturbed all layers with 50% of background values. Because the PM$_{2.5}$ above 1 km was on average smaller than 5 $\mu$m$^{-3}$, the perturbed ensemble spreads were really small. The surface observations affect the analysis fields, based on the calculated background error covariance. Therefore, small ensemble spreads above 1 km do not have an error correlation to the ground level. To clarify the perturbing procedure, we have revised the corresponding sentences (please, refer to **lines 159–161**).

**3. Page 6, line 172. Do you think that the static horizontal width of 100km and vertical width of 2km fit for all scenarios, for day and night? Any discussion about it.**

**Reply:** These horizontal and vertical ranges for the localization scales were sufficiently small so as to remove the spurious error correlation. Although 2 km seems too high during the nighttime, the vertical length scale in the **PH** matrix in Equation (5) is lower than 2 km during night. In other words, there is less vertical relationship between the localization scale and error covariances. We have attempted to explain more details about these localization scales in the revised manuscript (please, check out **lines 181–183**).

**4. Line 164. Same as above. Is the 30% standard deviation of LBC perturbation applied to the all layers?**

**Reply:** Similar to creating the initial ensemble, we applied this magnitude to all layers. We have added

this information to the revised manuscript (please, see **lines 171–172**).

**5. Section 2.2. The 3D-Var description in section 2.2 is too short, and needs to include more detail. What are the horizontal/vertical length scales, and model error covariance yielded by the NMC method? Could you show some plots about them?**

**Reply:** This study is a sort of comparison between the two DA methods. As we mentioned in the manuscript (lines 316–318), the comparison result in this study was not sophisticated so that it might not be a direct comparison. Actually, we are preparing an intensive comparison paper between the 3DVAR and two ensemble-based DA techniques (EnKF and EnSRF) with a more strict condition. Instead of adding extra plots in the revised manuscript, we have referred to a recently published paper (Lee et al., 2022), in which we introduced the development and application of the 3DVAR method, including the error covariances and length scales. Please, refer to Lee et al. (2022). We have added some more comments from Lee et al. (2022) into **lines 202–204**.

**6. Figure 4, it is better to include the corresponding 3D-Var increment for comparison.**

**Reply:** Following the comments #5 and #6, we have added an increment comparison for domain 1 in the supplementary information (please, see **lines 296–297 and Fig S2**).

**7. Section 3.3. Does the evaluation use the same observation data as those used in DA?**

**Reply:** Yes, we used the same observation data as those used in DA in the evaluations. Although the evaluations were not conducted with the spatially independent observations, we would say this is independent evaluations with respect to time because the statistical evaluations were carried out for each 1-day prediction (F00 to F24). We have specified the data used in Table 4 for readers' understanding (**lines 395–398**).

■ References

Volkamer, R., Jimenez, J. L., San Martini, F., Dzepina, K., Zhang, Q., Salcedo, D., Molina, L. T., Worsnop, D. R., and Molina, M. J.: Secondary organic aerosol formation from anthropogenic air pollution: Rapid and higher than expected, Geophys. Res. Lett., 33, L17811, 10.1029/2006GL026899, 2006.

Philip, S., Martin, R. V., Snider, G., Weagle, C. L., van Donkelaar, A., Brauer, M., Henze, D. K., Klimont, Z., Venkataraman, C., Guttikunda, S. K., and Zhang, Q.: Anthropogenic fugitive, combustion and industrial dust is a significant, underrepresented fine particulate matter source in global atmospheric models, Environ. Res. Lett., 12, 044018, 10.1088/1748-9326/aa65a4, 2017.

Jeong, J. I. and Park, R. J.: Influence of the Anthropogenic Fugitive, Combustion, and Industrial Dust on Winter Air Quality in East Asia, Atmosphere, 10, 790, 10.3390/atmos10120790, 2019.

Lee, S., Song, C. H., Han, K. M., Henze, D. K., Lee, K., Yu, J., Woo, J. H., Jung, J., Choi, Y., Saide, P. E., and Carmichael, G. R.: Impacts of uncertainties in emissions on aerosol data assimilation and short-term PM2.5 predictions over Northeast Asia, Atmos. Environ., 271, 11921, 10.1016/j.atmosenv.2021.118921, 2022.

---

## Author Comment (AC2)

**Response to Reviewer 2:**

First of all, thank you for your constructive comments on this manuscript. We have tried to address all the comments you made in the manuscript and also in the replies. The line numbers mentioned in the following replies are based on the revised manuscript with track changes.

■ General Comments

This manuscript describes the initialization of ground-level PM2.5 for a chemical transport model using the ensemble Kalman Filter (EnKF) method. Authors implemented EnKF in the CMAQ model and claim this method improves PM2.5 predictability compared to the 3DVar and No DA. The PM2.5 predictability of South Korea is further improved if the EnKF is applied not only to the nest domain but also mother domain with ground PM2.5 observation. The manuscript looks like to have a reasonable structure for the paper. It will be available for publication if it is improved with a minor revision.

■ Specific Comments

**1. In the manuscripts, ICs is an abbreviation for both initial conditions and initial concentrations. The same goes for BCs. Authors should only use the abbreviation "C" for either condition or concentration**

**Reply:** Thank you for this point. We first define the abbreviations of ICs and BCs (see **lines 17, 19, and 38)**, and then use both abbreviations in the revised manuscript.

**2. What are the background variables that are input to EnKF? I believe the EnKF uses background error covariance between PM2.5 and some meteorological variables. Is it correct that the experiments only update PM2.5, so other meteorological variables are the same as before DA? Adding a list of background and analysis variables is recommended.**

**Reply:** Yes, we updated only $PM_{2.5}$ via DA. Therefore, there were no error correlations between meteorological variables and $PM_{2.5}$. We used the identical meteorological conditions for all the DA experiments. Because $PM_{2.5}$ is a single control variable, we did not add a list of variables in the original manuscript. Regarding this point, we added same sentences into Section 2.1 (please, check out **lines 143–147)**.

**3. I am wondering about the observation operator for the PM2.5. Is PM2.5 one of the background variables? If not, the authors need to introduce the observation operator for it to calculate observation operator. Some descriptions for the observation operator would be better to be added in the manuscript.**

**Reply:** We agree with reviewer's comment that further descriptions of the observation operator are necessary. For a simple observation operator, we calculated $PM_{2.5}$ before the DA, using aerosol-related species via a post-processing tool in the CMAQ software package. This could also be possible because we used $PM_{2.5}$ as a single control variable. We have added same sentences about this process into the revised manuscript (please, refer to **lines 143–147)**.

**4. Line 174: What is the inflation parameter (alpha) for RTPS?**

**Reply:** Reply to this comment #4 is coming up with the following comment #5.

**5. Since you have described the parameter for the localization, it would be better to also describe the parameter for RTPS.**

**Reply:** We will try to response to comments #4 and #5 together. We set $\alpha$ of the inflation parameter to be 1.0. We here assumed that the meteorological model was perfect. Therefore, no perturbations were made for the ensemble spread. Another reason was that perturbing meteorological variables could also break dynamical balances in the model simulations. Throughout the experiments, we used 1.0 for the value of $\alpha$, and then inflated both the predicted ensemble (before DA) and assimilated ensemble (after DA) rather than using 1.2, the value used in Pagowski and Grell (2012) and Schwartz et al. (2014) (regarding this point, please refer to **lines 184–191**).

**6. Line 175: Why do you apply the RTPS to ensemble before and after DA? The RTPS compares ensemble spread before and after DA because the amount of inflation in RTPS is proportional to the ensemble spread reduced by the DA. So theoretically, RTPS can be applied only once after DA.**

**Reply:** Comments #4, #5, and #6 are a series of questions about similar issue. More specific details have been added into the revised manuscript (please, refer to **lines 184–191**).

**7. When you describe DA_icbc, can you show the pm2.5 field for domain1 which contains domain 2? For example, horizontal field of PM2.5 as in figure 4, but with domain 1. The distribution of PM2.5 over domain 1 can be clearer evidence showing the effect of boundary conditions.**

**Reply:** We agree with the reviewer's opinion. Instead of adding the $PM_{2.5}$ field for domain 1 to Fig. 4, we added **Fig. S5** into the supplementary information to describe the impacts of the updated (or increased) transboundary $PM_{2.5}$ by assimilating the concentrations from the ground stations in China. This additional figure could help readers to better understand the improved performances on 25 and 26 May, 2016 (high $PM_{2.5}$ episode during the KORUS-AQ campaign) in Fig. 8 (also, refer to **lines 374–376**).

■ Technical Comments

**8. Line 245: sate --> state**

**Reply:** We have changed it (please, see **line 266**).